# Targeted Therapy Against the Cell of Origin in Cutaneous Squamous Cell Carcinoma

**DOI:** 10.3390/ijms20092201

**Published:** 2019-05-05

**Authors:** Stephen J. Goldie, Ginevra Chincarini, Charbel Darido

**Affiliations:** 1College of Medicine and Public Health, Flinders University, Adelaide, SA 5001, Australia; stephen_goldie@hotmail.com; 2Peter MacCallum Cancer Centre, 305 Grattan St, Melbourne, VIC 3000, Australia; ginevra.chincarini@petermac.org; 3Sir Peter MacCallum Department of Oncology, The University of Melbourne, Parkville, VIC 3010, Australia

**Keywords:** Squamous cell carcinoma, cell of origin, differentiated cells, stem cells, cancer stem cells, therapeutic targets

## Abstract

Squamous cell carcinomas (SCC), including cutaneous SCCs, are by far the most frequent cancers in humans, accounting for 80% of all newly diagnosed malignancies worldwide. The old dogma that SCC develops exclusively from stem cells (SC) has now changed to include progenitors, transit-amplifying and differentiated short-lived cells. Accumulation of specific oncogenic mutations is required to induce SCC from each cell population. Whilst as fewer as one genetic hit is sufficient to induce SCC from a SC, multiple events are additionally required in more differentiated cells. Interestingly, the level of differentiation correlates with the number of transforming events required to induce a stem-like phenotype, a long-lived potential and a tumourigenic capacity in a progenitor, a transient amplifying or even in a terminally differentiated cell. Furthermore, it is well described that SCCs originating from different cells of origin differ not only in their squamous differentiation status but also in their malignant characteristics. This review summarises recent findings in cutaneous SCC and highlights transforming oncogenic events in specific cell populations. It underlines oncogenes that are restricted either to stem or differentiated cells, which could provide therapeutic target selectivity against heterogeneous SCC. This strategy may be applicable to SCC from different body locations, such as head and neck SCCs, which are currently still associated with poor survival outcomes.

## 1. Introduction

Skin squamous cell carcinoma (SCC) is one of the two most common non-melanoma skin cancers and occurs most frequently in sun-exposed regions of the skin and in immunocompromised patients. SCC of the skin poses differing burdens to health systems worldwide. Approximately 250,000 patients per year develop skin SCC in the United States [1]. In Australia, the estimated person-based incidence of SCC is thought to be 271 per 100,000 [2]. In neighbouring New Zealand, the highest incidence in the world is reported as 425 per 100,000 persons [3]. Patients with skin SCC may also develop multiple lesions and surgical resection can often be extensive and disfiguring, especially around cosmetically sensitive areas. Although this cancer is usually cured by surgical excision, approximately 8% of patients with skin SCC develop a recurrence and 5% present metastasis within 5 years. In patients with metastatic SCC, the prognosis is very poor, with only 10%–20% surviving more than 10 years [4,5]. There is therefore an urgent clinical need to prevent and treat skin SCC, and this requires a thorough understanding of the mechanisms that initiate this disease at the cellular level. 

## 2. Stem Cells in Cutaneous SCC

Epidermal keratinocytes are continually exposed to a wide range of environmental assaults including ultraviolet (UV) irradiation, chemical carcinogens, infection with human papilloma viruses (HPV) and other pathogens; and are therefore at a high risk of acquiring oncogenic mutations. The widespread exposure of the epidermis across sun-exposed body sites induces cellular transformation not only in a few cells but across the entire exposed field. However, relatively few skin cancers develop because most cells that acquire mutations are lost through the normal process of terminal differentiation [6], and it usually takes more than one genetic event for a tumour to develop. Most tumours are clonal in origin [7], meaning they can be traced back to the same single cell. It has been estimated that five events in humans, and two or three in rodents, are sufficient to transform a normal cell into a cancer cell [8]. Recent findings have shown that epidermal cells are more heterogeneous than previously thought with distinct proliferation and survival rates, varying long-term renewal and repair capacities and, evidently, differing oncogenic potential [9,10]. The most likely candidate population is the long-term epidermal resident cells, presumably stem cells (SC), which have an advantageous ability to result in tumour formation following few “genetic hits”; even though cells harbouring cancer-causing non-random mutations can maintain their physiological function until a positive selection of mutant drivers for clonal expansion occurs [11].

## 3. Heterogeneity of Cutaneous SCC

Skin SCCs exhibit complex cellular heterogeneity, with some cells expressing normal terminal differentiation markers while others are partially differentiated or undifferentiated. Depending on the oncogene or tumour suppressor involved and the cell of origin in which the oncogenic event took place, these SCCs manifest various differentiation states [12]. Furthermore, a functional heterogeneity has been established both in terms of clonal growth of SCC cells in vitro [13,14] and in xenograft models in immune compromised mice [15]. It was proposed that SCs should first transform into cancer stem cells (CSC) in order to induce SCC [16,17]. Clinicians and researchers have sought to identify unique markers of these CSCs, so that targeted therapies targeting those cells could form the “magic bullet” capable of preventing tumours from developing further, or even regressing or eradicating them. Whilst the likelihood that one target alone will provide benefit for patients is low, targeting multiple oncogenic mechanisms may have additive and/or synergistic effects and may therefore be ultimately effective in managing the disease.

A great deal of heterogeneity exists within the SC population, and this is supported by a vast publication list. Studies on head and neck SCC (HNSCC) have identified heterogeneity within tumours through differential expression of the CD44 cell surface glycoprotein, where cells expressing this factor show concomitant increased expression of the *BMI1* gene that has been shown to play roles in self-renewal and tumourigenesis [15]. Moreover, tumour heterogeneity can arise from additional non-genomic factors that lie within the microenvironment, including the availability of metabolites and signalling molecule gradients, which also contribute to the response of a tumour to specific drugs. This was shown when differential TGF-β signalling within SCCs influenced tumour drug responses; TGFβ in this context confers resistance to cisplatin, one of the most widely used anti-cancer drugs [18]. Although cisplatin treatment is highly effective for some skin SCCs, it remains to be seen whether combined anti-TGFβ/cisplatin therapy is beneficial for the whole spectrum of cutaneous SCC as well as for the treatment of HNSCC. Further identification of specific factors driving SCC pathogenesis from the cell of origin of these tumours will allow development of selective targeting approaches for better survival outcomes. 

## 4. The Cell of Origin in Cutaneous SCC

The well-established dual 7,12-dimethylbenz[a]anthracene (DMBA)/12-O-tetradecanoylphorbol 13-acetate (TPA) carcinogenesis protocol in mice demonstrates that cancer initiation is an irreversible event. Mice that are administered TPA one year after the last “priming” DMBA treatment develop cancer without significant delay in a similar fashion to mice that are immediately exposed to TPA after DMBA treatment [19]. The very short tumour latency, regardless of when TPA is applied, indicates that the underlying DMBA-induced cancer-initiating cells (CICs) are long-lived, slow-cycling cells that fail to disappear over time [20]. Whilst SCs have been proposed to be the CICs, emerging studies suggest that the acquisition of SC characteristics alone is insufficient for pre-malignant transformation and that oncogenic events occurring in a specific cell of origin are more relevant [12]. For example, high malignant capacity has previously been associated with the expression of mutant Harvey-Ras (*H-Ras*) in the hair follicle (HF), but when this mutation is expressed in the interfollicular epidermis (IFE), the lesions were rarely malignant, and only benign tumour growths known as papillomas form [21]. IFE tumour growth appears to be dependent on the continued administration of tumour promoters, having low potential to progress to malignancies in the context of Ras activation. Cells of the HF have additionally been implicated in SCC formation; however, the more lineage-restricted transit amplifying (TA) cell progeny lack the capacity to generate even benign papillomas within the same genetic context. Keratinocyte hyperproliferation and dedifferentiation are observed in response to expression of the *KRas^G12D^* mutant in HF SCs, where both MAPK and AKT-S6 signalling are increased, whilst *KRas^G12D^*-expressing TA cells are normal [22]. The absence of an abnormal phenotype in TA cells occurs due to the insufficiency of mutant oncogene expression required to generate long-term renewal potential, and thereby failing to initiate cancer development. These results imply that the type of cell in which specific oncogenes are activated determines the capacity for tumour development and can occur in progenitor, terminally differentiated or less differentiated TA cells.

SCC studies using *ShhCre^+^* mice expressing the *KRas^G12D^* mutant fail to initiate papilloma formation, in contrast to HF-specific K15Cre^+^ mice, where papillomas frequently manifest [23]. In addition, whilst SCCs were long believed to originate only from the HF bulge SCs, exciting studies proved that CICs could arise from the IFE population. Maintaining intact HF SCs whilst simultaneously removing the IFE SCs reduced the capacity to form papillomas and SCCs but did not abolish it following wounding and subsequent administration of the tumour promoter TPA [24]. Maintaining this capacity to form SCCs suggests that SCC-initiating cells are from slow-cycling populations, not only from the HF but also residing in the IFE [20,24].

Conversion to malignancy, however, can be established through the concurrent loss of the tumour suppressor gene *p53* in *KRas^G12D^* expressing HF SCs [22], although gain-of-function of *p53^R172H^* in this context confers a poorer prognosis [25]. Both carcinogen- and genetically-induced mouse skin SCCs show recurrent mutations in Ras family members with copy number alterations in the *p53* gene [26]. While hyperplasia and markers of epithelial-mesenchymal transition (EMT) are evident in cells lacking *p53*, SCCs in this model only emerge following the induction of this second ‘hit’. Furthermore, expression of both *SmoM2* and *p53* in InvCre-ER positive is required for tumour development, while the expression of *SmoM2* alone in SCs is sufficient for tumour development [27]. Therefore, the nature of the CIC and the specific oncogene involved, in addition to the contribution of the cellular microenvironment, are the main drivers of cancer progression, characterise the resulting tumour type and define its malignant potential.

The concept of “a CIC transforming into a CSC” in SCC was proposed by Patel et al. [28]. By simply sorting SCC cell lines based on CD133 expression, the authors were able to show that CD133^+^ CICs recapitulate heterogeneous SCCs in xenograft models. The CD133^–^ cells were unable to maintain SCC growth in serial transplantation studies [28]. Siegle et al. also attempted to elucidate unique identifying genetic markers in SCC CICs [29]. The authors compared gene expression of mouse samples from cutaneous SCC CICs to HF SCs and epidermal progenitor cells. They found differential expression of a handful of transcription factors, most notably SOX2. By immune staining, they confirmed SOX2 is absent in normal skin samples and only present in SCC. The authors show Sox2 is expressed in undifferentiated cells at the tumour/stroma interface that are also positive for basal-like cell markers α6 and β1 integrin [29], which might be considered the “tumour stem cell compartment”. The same pattern was seen in human skin and SCC samples. Xenograft studies using stable and inducible knockdown of *Sox2* showed a reduction in the growth of tumours, which highlights the critical role of SOX2 in the development and maintenance of SCC [29]. These data correlate well with work published previously by the Blanpain lab showing increased expression of SOX2 in the spectrum of sun-damaged skin, ranging from actinic keratosis to SCC [30,31]. Together, these findings point to SCC potency in a manner that is dependent on the cell-specific expression of oncogenes (Figure 1). 

## 5. Evaluation of Normal Stem Cell Markers in SCC

Presenting evidence for the “Keratinocyte Stem Cell (KSC)” as the likely cell of origin in cutaneous neoplasias, Lotti et al. investigated the role of Survivin in the development of cutaneous SCC [32]. The IAP protein Survivin had previously been identified as a marker of KSCs. In cell culture experiments, siRNA knockdown of Survivin reduced colony formation and cell viability compared to control cells. It also had deleterious effects on stem-like properties such as rapid adherence and expression of stemness-associated markers OCT-4, NOTCH1, CD133 and β1-integrin and increased expression of differentiation markers K10 and Involucrin. In 3D reconstruction skin models, Survivin knockdown cells developed an epidermis as thick as controls and showed reduced ability to form SCCs and invade into the dermis. The authors present IHC data to correlate the reduced proliferation of Survivin-depleted cells with reduced Ki67 staining and the lack of invasiveness with a lower expression of metalloproteinase-9 (MMP9). This reduced “aggressive” potential of the Survivin siRNA SCCs is reflected in the lower expression of Psoriasin, a known marker of poor prognosis in epithelial cancers. Normal keratinocytes transduced with retroviral vectors expressing oncogenic *RAS*, and the inhibitor of NFκB (*IκBαM*) promoted the expression of Survivin. Whilst additional knockdown of Survivin in those cells reduced proliferation, it could not prevent tumour formation in xenograft models, perhaps only delaying their development. Unfortunately, experimental numbers were too low to draw significant conclusions from these data. Furthermore, the authors analysed their tumour samples for the cancer-associated markers HIF-1α, VEGF and CD51 (αv-integrin) and showed they are reduced in Survivin siRNA tumours, inferring from this that the overexpression of Survivin seen in SCC is the responsible factor for the poor prognosis of these lesions.

Following the investigation of the role of Survivin in cutaneous SCC development, overexpression of Survivin was observed across several tissue types of SCCs. Santarelli et al. reported on the role of Survivin as a prognostic factor in oral SCC [33]. By IHC staining of oral SCC tumour samples, a significant correlation between the larger T-stage tumours and Survivin expression was demonstrated. The increasing strength of nuclear Survivin staining correlated negatively with patient survival in their study population. The same group have also reviewed the potential to target Survivin in the management of SCC in multiple tissue types, using a variety of druggable mechanisms and pathways to prevent the anti-apoptotic effect of elevated Survivin [34]. 

When single cell gene expression profiling was used to identify markers of IFE SCs in cultured human epidermal cells, FERM domain containing 4A (FRMD4A) was one highly predicted SC marker [35]. Furthermore, 14 potential epidermal SC markers were examined by Q-PCR on SCC cells, and interestingly, *FRMD4A* was the only gene that was consistently highly upregulated in SCC, confirming it as a candidate SCC SC marker [13]. FRMD4A was also reported to connect the Par-3 complex to the Arf6 guanine-nucleotide exchange factor and regulate the assembly of adherens junctions in cultured epithelial cells [36]. This SCC SC marker is of considerable interest for several reasons. First, examining FRMD4A expression provides a means to explore the relationship between markers of normal SCs and CSCs. Second, FRMD4A is reported to regulate intercellular adhesion and maintain E-cadherin expression in normal cells [36], despite the prevailing dogma that E-cad-mediated adhesion is downregulated in invasive epithelial cancers [37,38]. FRMD4A expression is found exclusively in the basal layer of the human epidermis and is lost upon differentiation of keratinocytes. Stable knock-down of *FRMD4A* in human SCC cell lines reduced both colony-forming efficiency and overall growth potential in in vitro and in vivo xenograft models. Data mining of databases of HNSCC and other cancers reveals that high FRMD4A expression levels predict poor prognosis [39]. The overall survival of mice xenografted with human SCC was improved by inducible knockdown of *FRMD4A*, confirming the prognostic value identified in HNSCC [39]. Mechanistically, the protein product of FRMD4A is an Ezrin Radixin Moesin (ERM) protein, and like other ERM family members, reducing expression of FRMD4A appears to modulate the Hippo pathway, which has been implicated in the control of tissue growth and cancer development [39]. These data suggest that in a normal stem cell context, FRMD4A interacts with the Par-3 polarity complex to regulate cell adhesion. Upregulation of FRMD4A and its mislocalisation to the nucleus are associated with a CSC phenotype and SCC development.

## 6. Pro-inflammatory Secreted Factors in SCC Development

Arwert et al. investigated the potential role of inflammation driving increased proliferation and tumourigenesis in skin [40]. In InvEE transgenic mice, overexpression of activated MAPK kinase 1 (MEK1) in the suprabasal non-dividing differentiated cell layers (*InvEE* transgenics), results in epidermal hyperproliferation and skin inflammation. Interestingly, wounding of these animals leads to the reproducible development of papillomas at the site of injury [40]. The authors found that inflammation, and in particular secretion of cytokine IL-1α, was key to promoting tumour development by engaging hyperproliferative mechanisms within the basal layer. Growth of papillomas could be reduced by treating wounded animals with the non-specific immune-suppressing drugs dexamethasone and cyclosporin. The same study proved the specificity of the IL-1α mechanism by treating the same animals with an IL-1 receptor antagonist [40]. 

Another example is Lrig1, a SC marker within the IFE and an antagonist of EGFR [41]. Recently, the link between the role of inflammation-driving carcinogenesis and this specific *Lrig1*-positive SC population has been established in SCC. Chen et al. studied the interplay between IL-17R and the EGFR pathway to show that Lrig1 functions as a tumour suppressor and its expression is reduced in human SCC lines [41]. Using their sophisticated transgenic mouse model, the authors identified IL-17R as playing an essential role in wound healing and tumorigenesis in *Lrig1*-positive SCs. Their data showed that IL-17R is responsible for the mobilization of *Lrig1*-positive SCs in wound healing experiments and in tumours in response to wounding [41].

Similar findings were established in the context of Grainyhead-like 3 (GRHL3), a conserved developmental transcription factor, which is essential for epidermal differentiation and barrier formation [42,43,44]. *Grhl3*-null mice die shortly after birth due to excessive water loss, a result of compromised skin barrier formation [45,46]. GRHL3 also plays an important role in tumour suppression. Conditional epithelial-specific deletion of *Grhl3* in adult mice induced spontaneous skin tumours as the animals were aged and showed increased susceptibility to DMBA/TPA chemical-induced SCC [47,48,49,50]. *Grhl3* is normally highly expressed in the suprabasal layer, and the mechanism whereby tumours in the stem cell/basal cell compartments were initiated following loss of *Grhl3* was until recently poorly understood. Cell culture conditioned media from *GRHL3*-deficient HaCaT cells increased the proliferation of control HaCaT cells in vitro [51]. Cytokine arrays identified only one factor significantly elevated in the conditioned media, the Thymus and activation-regulated chemokine (TARC), also known as CC-motif ligand 17 (CCL17), which promotes Th2 lymphocyte recruitment and epidermal hyperplasia. The literature suggests that TARC plays a role in several inflammatory skin diseases and is significantly associated with multiple malignancies [52,53]. Interestingly, *Grhl3*-null epidermis showed increased inflammatory cells and activation of the STAT3 signalling pathway. Furthermore, treatment of *Grhl3*-null mouse skin with an inhibitor of TARC in an ex-vivo model, demonstrated a partial rescue of the hyperproliferative phenotype and barrier breakdown [51].

Non-coding microRNAs (miRs), which have been shown to play a role in the dysregulation of cell behaviour, tumour progression and even metastasis, can also regulate the secreted factors in SCC development [54]. Toll et al. compared miRs expression in normal skin compared to sun-damaged skin and SCC by gene expression microarray [55]. While many miRs were upregulated, the authors identified *miR-204* as the only miR significantly downregulated in their SCC screen and confirmed this by Q-PCR. They next silenced *miR-204* in human SCC cell lines and found a mixed picture, suggesting that DNA methylation is only one of multiple mechanisms resulting in *miR-204* downregulation. The potential mechanism by which loss of *miR-204* could drive SCC development was shown to be upstream of the FGF-STAT3 signalling pathway [55]. STAT3 activation seems to be the most frequent mechanism and has been reported in various mouse models of SCC [56]. Overall, loss of *miR-204* initiated a STAT3-driven pro-inflammatory response in sun-damaged skin cells leading to the development of SCC.

## 7. Genome Wide Association Studies and Cutaneous SCC

Given the high incidence of cutaneous SCC and the availability of large numbers of tissue samples, it is perhaps surprising that there have been relatively few genome wide association studies to investigate common genetic drivers of the disease. One such study carried out by Chahal et al. identified 11 susceptibility loci of interest [57]. The authors describe these loci being associated with pigmentation genes, as would be anticipated with the UV radiation aetiology of cutaneous SCC. The Watt Lab took a different strategy to explore the genetic interplays occurring in SCC and SCs [58]. Using a library of shRNAs, they performed high throughput RNAi screens on normal human keratinocytes and a well-established human cutaneous SCC line. The bioinformatic analysis of the functional outcomes of these cells in culture allowed the identification of YAP1, a transcriptional co-activator known to be a key downstream effector of the Hippo pathway. The YAP1 paralogue TAZ also regulates the growth of SCC. Further analysis of gene targets of the transcription factor TEAD elucidated potential mechanisms, whereby the Hippo pathway and expression of YAP1 may influence the behaviour of SCs in the normal epithelium and in SCC (Table 1).

## 8. Targeted Therapy and the Cell of Origin in Cutaneous SCC

The central concept of targeted therapy, also called precision oncology, requires additional testing to determine features of a patient’s malignant disease that may inform treatment decisions. Molecular assays have formed part of the standard diagnostic work up for various malignancies, such as estrogen receptor staining in breast cancer or mismatch repair immunohistochemistry in colorectal carcinoma. These assays attempt to define the natural history of the malignancy as well as context-specific markers in order to guide the most appropriate treatment. Genomic biomarkers paired with targeted therapies have proved highly efficacious in some cases, such as *HER2* amplification and trastuzumab in breast cancer [59], *BRAF* mutation and combined BRAF and MEK inhibition in melanoma [60,61] and *EML4*-*ALK* fusions and crizotinib in lung adenocarcinoma [62]. These successful examples have set the benchmark for precision oncology and prompted the characterisation of genomic biomarkers for all malignant diseases, including cutaneous SCC. Furthermore, meta-analyses comparing personalised and non-personalised treatments have demonstrated that targeted therapies are associated with a higher response rate and better progression free survival [63,64].

Comparative analysis of pathway alterations and global cancer functional event (CFE) signatures have shown concordance in cell lines and cutaneous SCC tumours [65]. Gene expression profiling has been less commonly employed in targeted therapy as a stand-alone assay. Interestingly, a large proportion of the variance in gene expression between tumours is explained by tissue of origin [66]. From in vitro high throughput drug screens paired with comprehensive genomic profiling of cell lines, gene expression has proven to be the most useful feature to predict drug sensitivity [65,66]. It is important to note that tissue of origin is almost as good a predictor as gene expression and that the two are highly correlated [65]. When considering predictors of drug sensitivity within a given tumour type, genomic features (either driver mutations or copy number alterations) coupled with gene expression-based profiles could accurately predict drug sensitivity [65]. Whilst these results are highly dependent on the background prevalence of genomic features in the dataset in question, they highlight that multi-omics approaches incorporating multiple data types may add value over unilateral approaches. This was shown in human tumour datasets, where integrated approaches leveraging both transcriptomic and genomic data have shown promising results in predicting drug sensitivity [67,68]. Therefore, targeting specific factors that contribute to the tumourigenesis of the cell of origin could add another selectivity level to targeted therapy. Dedifferentiation of a differentiated cell through the expression of secreted factors (e.g., TARC/CCL17) or activation of specific oncogenes in SCs (e.g., FRMD4A) are key targets against the CIC. 

## 9. Conclusions

Identifying novel targets that drive SCC genesis is imperative given the increasing incidence of this disease and will underwrite the development of targeted therapies, especially for high-risk patients. Identification of specific markers of the transformed cell of origin would therefore be a primary goal, enabling dissection of the mechanisms behind its development and progression. Furthermore, biological insights into cutaneous SCC would provide a basis for comprehending more aggressive and heterogeneous epidermal cancers, such as HNSCC. Future studies will ultimately establish a molecular signature, whereby patients can be stratified to receive drugs that directly target their cell of origin-dependent oncogenic drivers, reflected by the tumour differentiation status, and potentially bypassing drug resistance in response to therapy. The tissue of origin and the cell of origin will dictate responses to targeted therapies. Personalised inhibitors will determine whether selective targeted therapy against CIC-dependent factors is sufficient to eradicate heterogeneous SCC, manage the disease and improve patient quality of life and overall survival. 

## Figures and Tables

**Figure 1 ijms-20-02201-f001:**
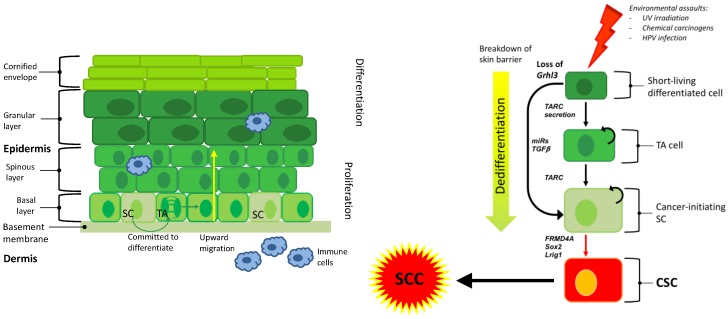
Exposure to environmental assaults transform SC or dedifferentiate a differentiated cell through the expression of secreted factors or activation of specific oncogenes. This cell becomes a CIC that give rise to a CSC and ultimately to SCC. Transit amplifying (TA), stem cell (SC), cancer stem cell (CSC), squamous cell carcinoma (SCC).

**Table 1 ijms-20-02201-t001:** Summary of factors that are shown to regulate tumour-initiating cell growth in cutaneous SCC.

Factor	Potential Mechanism	References
**CD44**	Cell surface marker elevated in subpopulations of SCC, i.e., putative tumour stem cell marker	[15]
**CD133**	Putative tumour stem cell marker in SCC	[28]
**TGF-β**	Promotes chemotherapy resistance in SCC	[18]
**BMI1**	Tumour self-renewal and tumourigenesis	[15]
**Sox2**	Putative tumour stem cell marker in SCC	[29]
**FRMD4A**	Putative tumour stem cell marker in SCC; interacts with Hippo pathway	[39]
**YAP1**	Key downstream effector of the Hippo pathway, influencing the behaviour of stem cells in the normal epithelium and in SCC	[58]
**Lrig1 (loss)**	Putative tumour stem cell marker in SCC and tumour suppressor.	[41]
**Grhl3 (loss)**	Conserved developmental transcription factor, essential for epidermal differentiation, barrier formation and SCC initiation	[51]
**TARC/CCL17**	Cytokine driving hyperproliferation in Grhl3 deletion model	[51]
**IL-1α**	Cytokine driving papilloma formation in MEK1 overexpression model	[40]
**IL-17R**	Cytokine implicated in the mobilization of Lrig1-positive stem cells in wound healing experiments, and in tumorigenesis	[41]
**Survivin**	Overexpression of Survivin promotes the development of SCC in multiple tissue types and results in poor prognosis	[32]
**miR-204 (loss)**	Loss of miR-204 drives STAT3 pro-inflammatory response in sun-damaged skin leading to the development of SCC.	[55]
**MEK1**	Overexpression promotes hyperproliferation and skin inflammation, leading to papilloma formation	[40]

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
