# Peer review of "Targeted Therapy Against the Cell of Origin in Cutaneous Squamous Cell Carcinoma"

_ijms, 2019, doi:10.3390/ijms20092201_

Round 1
Reviewer 1 Report
The paper discusses the targeted therapy against skin SCC by reviewing the most recent findings in the field. The paper is well organized, scientifically sound and well written. In my opinion, the paper brings a high level of methodical literature survey and can be published in its current form. I would only add the term "mini-review" in the title, considering the lenght of the article, 6 pages, and the relatively limited number of references.
Author Response
We thank you for the thoughtful suggestions. 1.5 pages and additional references are added to further support the publication.
Reviewer 2 Report
The manuscript entitled “Targeted therapy against the cell of origin in cutaneous squamous cell carcinoma” is a well-written review that describe the markers related to stem cells (SC) in cutaneous squamous cell carcinoma (SCC). This manuscript covers a variety of actual developments in identification of molecular signature of SCC-SC. I suggest adding a brief paragraph describing the possible role of Survivin in cutaneous SCC-SC cell growth. Indeed, some studies indicate Survivin as a key gene in regulating SCC cancer stem cell formation and cutaneous SCC development [1]. Furthermore, several studies demonstrated the potential of Survivin in Head and Neck SCC (HNSCC) as prognostic marker and therapeutic target for anticancer treatment [2, 3].
[1]: Lotti R, et al. Survivin Modulates Squamous Cell Carcinoma-Derived Stem-Like Cell Proliferation, Viability and Tumor Formation in Vivo. Int J Mol Sci. 2016 Jan 12;17(1). pii: E89. doi: 10.3390/ijms17010089.
[2]: Santarelli, A, et al. Nuclear Survivin as a Prognostic Factor in Squamous-Cell Carcinoma of the Oral Cavity. Appl Immunohistochem Mol Morphol. 2017;25(8):566-570.
[3]: Santarelli A, et al. Survivin-Based Treatment Strategies for Squamous Cell Carcinoma. Int J Mol Sci. 2018 Mar 24;19(4). pii: E971. doi: 10.3390/ijms19040971.
Author Response
We thank you and the Reviewers for the thoughtful suggestions and minor comments. Please find attached our revised review entitled, “Targeted therapy against the cell of origin in cutaneous squamous cell carcinoma”, including a section on Survivin, for consideration by IJMS.
In this version, we discuss the role of Survivin in the cell of origin of SCC and how it impacts on patients’ prognosis as suggested by Reviewer #2.